# A standardised protocol for measuring farmland biodiversity outcomes across European Farmer Cluster landscapes

Rachel N. Nichols[1]*, Graham S. Begg[2], Lisette Cantú-Salazar[3], John M. Holland[1], Youri Martin[3], John Tzilivakis[4], Sarah Vray[3], Douglas J. Warner[4], Alon Zuta[2], Niamh M. McHugh[1]

1 Farmland Ecology Unit, Game and Wildlife Conservation Trust, Fordingbridge, Hampshire, United Kingdom, 2 Ecological Sciences Department, The James Hutton Institute, Invergowrie, Dundee, United Kingdom, 3 Biodiversity Monitoring and Assessment group, Environmental Sensing and Modelling unit, Luxembourg Institute of Science and Technology, Belvaux, Luxembourg, 4 Agriculture and Environment Research Unit (AERU), School of Health, Medicine and Life Sciences, University of Hertfordshire, Hatfield, Hertfordshire, United Kingdom

* rnichols@gwct.org.uk

## Abstract

Farmland biodiversity has declined, with agricultural intensification cited as a major contributing factor. Therefore, monitoring farmland biodiversity in a consistent manner is a crucial aspect of determining if conservation efforts are aiding in its recovery. "Farmer Clusters" comprise groups of neighbouring farmers who have identified and instigated their own biodiversity targets as a collective. These provide an opportunity to assess the impact of collaborative conservation efforts on farmland biodiversity at the landscape-scale. Through the FRAMEwork project (Farmer clusters for Realising Agrobiodiversity Management across Ecosystem), an EU Horizon 2020 project, eleven Farmer Clusters were established across Europe, and their biodiversity monitored in a Before-After-Control-Impact (BACI) experiment. The aim of this paper is to describe and critically evaluate the methods applied to monitor biodiversity across these landscapes. Once relevant landscape-scale biodiversity indicators were identified, the protocol was assembled. Guidance to select appropriate survey squares and transects in different farming systems and European landscapes that are representative of the different farmland habitats are provided. The monitoring protocol also describes how to conduct biodiversity surveys of birds, pollinators (bumblebees, butterflies, solitary bees, and hoverflies), and vegetation within these survey squares. The methods comprise of a combination of well-documented procedures, presented as a single, standardised protocol that can be replicated in different farming systems and landscapes throughout Europe. Anticipated results are also presented, demonstrating how the methods might contribute towards the BACI analysis, as well as multi-taxa, community-level biodiversity analysis. Finally, an evaluation of the biodiversity indicators and survey methods used is also given. Through FRAMEwork,

**Data availability statement:** All relevant data are within the manuscript and its Supporting information files.

**Funding:** This Project has received funding from the European Union's Horizon 2020 research and innovation programme under grant agreement No 862731. The funders had no role in study design, data collection and analysis, decision to publish, or preparation of the manuscript.

**Competing interests:** The authors have declared that no competing interests exist.

these methods will be used to assess the effectiveness of Farmer Clusters at improving landscape-scale biodiversity.

## Introduction

As the global human population is expected to reach 9.7 billion by 2050 [1], global food security and the sustainability of food production has become a pressing issue [2]. Improved agricultural sustainability promotes greater emphasis on reducing inputs and utilising ecosystem services and functions, often through the conservation of farmland biodiversity. Higher levels of biodiversity have been linked to improved pollination services [3–5], pest management [6–9], nitrogen fixation [10,11] and carbon capture [12]. Improving farmland biodiversity is therefore key when maintaining a sustainable level of agricultural output.

Agri-environment schemes (AES) were first introduced in Europe in the 1980s as a way to improve farmland biodiversity. They were formally incorporated into European Union (EU) policy in 1992 with the reform of the Common Agriculture Policy (CAP). This policy shift marked the beginning of a new environmental focus of the CAP in response to biodiversity loss, soil erosion and the decline of traditional agricultural landscapes [13]. Through AES, farmers are incentivised to adopt more sustainable and environmentally friendly farming practices. However, throughout Europe the implementation of AES tends to follow a 'top-down' approach where the environmental priorities are set by government agencies, and farmers must follow a set of criteria to be eligible for schemes.

It is generally unclear how AES are performing. Although some AESs are formally monitored by national governing bodies (e.g., in England by Natural England; [14]), and there is the capacity to monitor the multi-taxa success of AESs across the landscape through government-funded projects [15,16], there is no Europe-wide standardised biodiversity monitoring protocol that takes the implementation of biodiversity-friendly activities, such as AESs, into consideration. Moreover, individual farmers are not provided with feedback on the wider successes or failures of the actions they have instigated for each AES option, unless they hire an environmental advisor to conduct individual assessments and monitoring. Additionally, due to the way in which a large proportion of AESs have typically been taken up, by individual farm businesses as per the application process (though applications are submitted jointly in the Netherlands [17]), the potential to truly alter the landscape through coordination is missing. Thus, not only is it unclear if individually placed AES options are achieving their aims, but there is also no system in place to amend and improve an individual AES in situ on a farm, and there is little-to-no communication between farms on what actions have been put in place and where. This is where Farmer Clusters (FCs) have the potential to improve the current system.

Farmer Clusters are a UK initiative where communities of farmers, located in the same region, share knowledge, and support and motivate each other to make environmental improvements on their farms, with the assistance of a 'facilitator', someone who is an experienced farm advisor, knowledgeable in AESs and farming practices

[18]. By selecting their own environmental targets and instigating tailored AES or biodiversity-friendly farming practices with the assistance of the facilitator, FCs are expected to be more invested in their farm's biodiversity, and work collectively as a group to improve biodiversity across their landscape. However, it is still unclear if FCs are delivering broader benefits for farmland biodiversity beyond their targeted species or environmental management aims.

Although several large multi-taxa programmes exist with well-developed protocols aimed at monitoring biodiversity [15,16,19], more recently also considering farmer collaboration across the landscape [20], none were designed specifically to operate at the scale of the FC or the diverse landscape sizes that they encompass. There was a need to identify indicator species that were relevant to the landscape and aligned with the interests of the FCs, as well as a protocol adaptable across different European countries and farming systems.

The FRAMEwork project (Farmer clusters for Realising Agrobiodiversity Management across Ecosystems), a multi-partner project funded by the EU Horizon 2020 initiative, was established to investigate whether the FC approach could be employed across Europe. A key aspect of FRAMEwork was the development and testing of methods and tools for the monitoring and evaluation of farmland biodiversity, a main criterion for assessing the performance of the FCs. The overarching aim was to determine if biodiversity could be improved overall across the landscape by the FC selecting their own biodiversity targets and biodiversity-friendly farming initiatives. To test this, a Before-After-Control-Impact (BACI) experiment was designed, using a control for ten out of the eleven FCs.

The aim of this paper is to a) describe the FRAMEwork project's methods to select and survey landscape-scale biodiversity indicators, and b) evaluate the appropriateness of the selected taxonomic indicators and the methodologies for data collection. However, the described protocols are also intended to be applicable for any future FC facilitators or practitioners to monitor their FC biodiversity from inception.

## Methods

### The FRAMEwork Farmer Clusters

To understand how landscape management through FCs could impact biodiversity, it was important that FRAMEwork represented a variety of European farming systems and landscapes. Through collaboration with multiple project partners, eleven FCs across nine European countries were successfully established (Fig 1A). Landscapes and farming systems identified included lowland valleys, mountainous pastureland, orchards and arable plains (Table 1). Each FC focused on its own environmental priorities, which ranged from improving the overall diversity of plants, birds, and pollinators, to actions such as planting hedgerows and protecting soil health, and even conserving specific species such as the red-legged partridge (*Alectoris rufa*) and barn owl (*Tyto alba*) [18].

It was hypothesised that the targeted biodiversity-friendly farming activities, initiated collaboratively by the FC, would lead to measurable improvements in overall farmland biodiversity which would be captured through standardised monitoring. Therefore, to determine the effect of FCs on overall farmland biodiversity, a BACI experiment was designed. Each FC was paired with a comparable control in a nearby, similar landscape and farming system (approximately 2 km away; Fig 1B). This approach allows the assessment of change in biodiversity over time by comparing trends in intervention and control sites before and after FC biodiversity measures were implemented. Before the project commenced, a formula ([21]; S1 Appendix) was used to calculate that a minimum of four treatment (FC) squares and four control squares were required to be 72.8% confident of correctly predicting the direction of biodiversity change on control versus FC site by the end of the project in a BACI analysis. Each biodiversity indicator will be assessed individually, and separately for each FC, through generalised linear mixed effects models that test the interaction effect of time (before or after) and treatment (FC or control; [22]). Due to the close proximity of other FCs to the English FC, selecting a control site near the FC was not feasible [23], and instead, between 10–20 survey squares were selected for biodiversity monitoring, providing 67–70% confidence of correctly predicting the direction of biodiversity change [21].

**A**

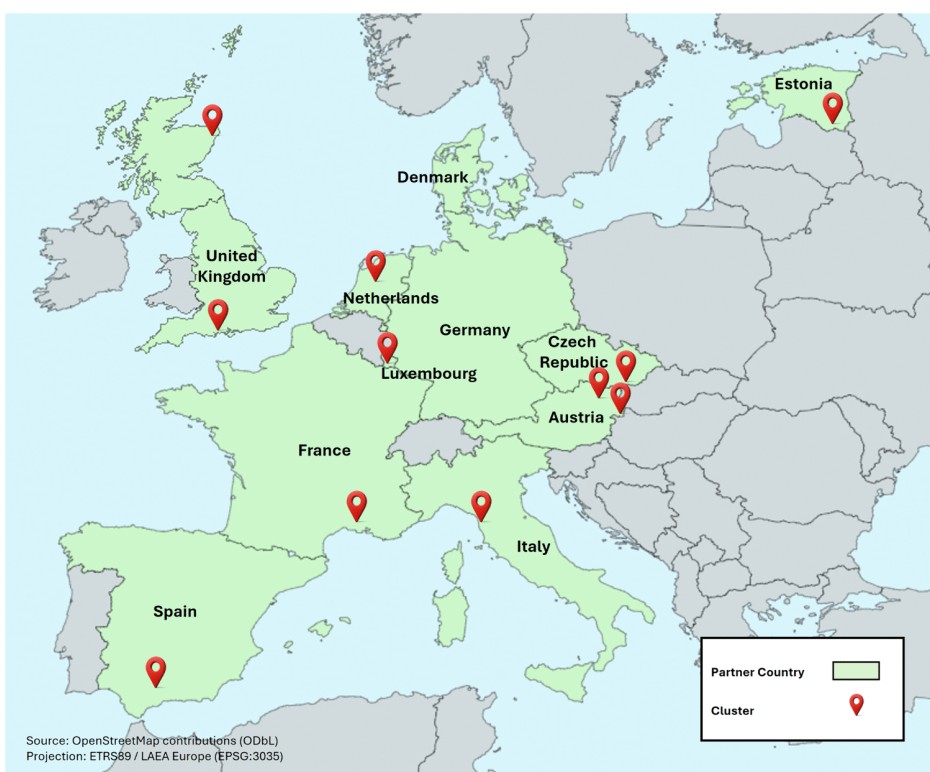

**B**

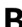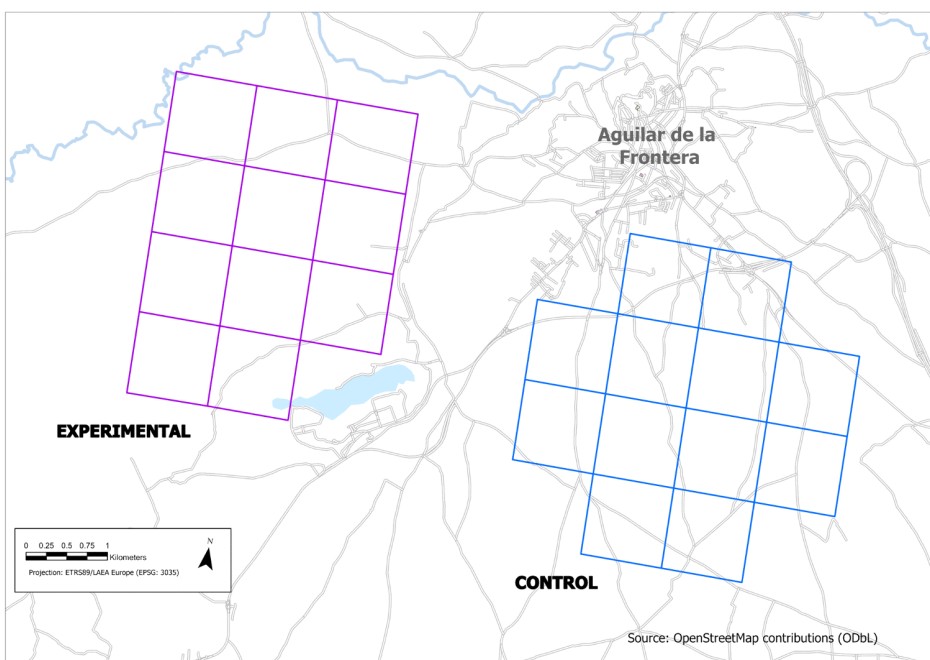

**Fig 1. Farmer Clusters in the FRAMEwork project. (A)** Locations of all eleven Farmer Clusters and the control sites across Europe that are part of the FRAMEwork project (Cluster marker), and all FRAMEwork project partner counties in green. **(B)** A demonstration of the relative position of the Farmer Cluster (experimental) and control survey squares of the Spanish Farmer Cluster.

**Table 1. Farmer Cluster locations and details.**

| Country | Local area | Climate | Farming system | Landscape |
|---|---|---|---|---|
| Austria | Burgenland | Cool Temperate Dry | Mixed | Plains, rolling hills |
| Austria | Mostviertel | Cool Temperate Moist | Cattle farms | Alpine foothills |
| Czech Republic | Velké Hostèrádky | Cool Temperate Moist | Mixed | Rolling hills |
| England | Cranborne Chase | Cool Temperate Moist | Arable/Mixed | Rolling hills |
| Estonia | Kanepi kihlkund | Cool Temperate Moist | Arable/Mixed | Dome landscape |
| France | Basse-Durance | Warm Temperate Dry | Orchards | Plains, Durance Valley |
| Italy | Val Graziosa | Warm Temperate Moist | Olive grove | Lower mountain |
| Luxembourg | Born | Cool Temperate Moist | Orchards | Rolling hills, valley |
| Netherlands | Zeeasterweg | Cool Temperate Moist | Arable | Lowlands |
| Scotland | Buchan | Cool Temperate Moist | Mixed | Rolling hills |
| Spain | Cazadores de Aguilar | Warm Temperate Dry | Olive grove | Dry plains |

## Biodiversity monitoring

**Biodiversity indicators.** Identifying biodiversity indicators that communicated information regarding the status of the environment across the FC landscapes was an important step in the FRAMEwork project. These biodiversity indicators were not necessarily the target species/taxa of all the FCs, but it was intended that they would be proxies by which the overall biodiversity of each FC could be assessed over the course of the project. An initial list of indicators was drawn up by the project partners through an extensive literature review, these were then narrowed down through a rigorous evaluation process and constant feedback loop with project partners, facilitators, and biodiversity experts in a step-by-step elimination process to determine the final indicators for the project. Step 1: many indicators were identified from similar large-scale farmland biodiversity and pan-European projects such as Biodiversity indicators for organic and low-input farming systems (BioBio; [24], Farm4Bio [25], Quantification of Ecosystem Services for Sustainable Agriculture (QuESSA; [26]), and European Monitoring of Biodiversity in Agricultural Landscapes (EMBAL; [19]), as well as additional indicators that were of interest to project partners (full list in S2 Appendix). The data collection methods and associated difficulty levels for monitoring each were also noted (S2 Appendix). Step 2: each indicator was evaluated for its relevance to the FRAMEwork project before being selected for the next stage, i.e., its ability to be a proxy for biodiversity and inform at the FC landscape-scale. Step 3: all indicators were then assessed for their interest to multiple project partners (together with those not selected in step 2), including if project partners already had skills in identification or monitoring of the indicator, and brought forward to the next step. Step 4: it was agreed that a maximum of six indicators representing multiple categories (Kingdoms/Classes) would both suffice and be achievable within the timeframe for data collection, and the final list was defined, including the taxonomic rank to which all FCs would conduct monitoring. This resulting list of six biodiversity indicators to be monitored across all FCs included: flowering plants (to genus) and other vegetation features (including groups of species, percentage cover, colour diversity of flowers etc.); butterflies (identified to species level); bumblebees (to species or morphospecies level); solitary bees (to family, or genus and species where skills allowed); hoverflies (at the group level); and farmland birds (to species level).

Methods for monitoring biodiversity (outlined below) were determined jointly between biodiversity experts within FRAMEwork. Previously well-documented methods, altered where necessary to match the requirements of the FRAMEwork project, were combined into a single, standardised protocol that was applicable across all FCs. After FCs were initially established, biodiversity surveys were conducted in 2021 (trial surveys) and 2022 (biodiversity baseline survey), before any additional biodiversity sensitive measures were put in place. All FCs were surveyed again in 2024 to quantify the initial impacts of FC management on overall biodiversity in a BACI design. Due to the high densities of FCs in the

South of England the English FC lacked a control area for comparisons. Surveys were therefore conducted annually from 2021–2024, with the aim of examining changes in biodiversity over time in the FC. It is anticipated that any new or established FCs looking to monitor their biodiversity would consider biodiversity changes over time, rather than also looking for a control site to conduct a BACI analysis.

**Survey design.** To ensure that biodiversity monitoring was representative of the FC landscape, a spatially balanced design was used. Each FC (and its control) was divided into squares scaled towards its average farm size, and surveys were conducted in a stratified sampling manner. Each square was typically 1 km$^2$, but smaller squares of 0.25 km$^2$ (500 m x 500 m) were used in some FCs where farms were small or fragmented. In each FC, a subset of these squares (at least four) and a comparable number of control squares were surveyed. The control squares were selected to be as similar as possible to the FC squares in terms of farm characteristics: farming system, landscape, and habitat composition. Where FCs were larger or more aggregated, a 1 km grid was placed over the FC area (Fig 2A). Any squares that contained land outside the FC, or had less than 75% agricultural land coverage, were excluded. For smaller or more fragmented FCs, squares were selected to maximise the overlap with the farms while maintaining the required minimum spacing of bird transects (typically 500m and no closer than 200m between any points). Each survey square had a breeding bird transect, multiple corresponding pollinator and vegetation transects (co-locations; Fig 2B and 2C), and within squares distinct habitats were identified and mapped (Fig 2C). Each co-location consisted of a vegetation survey transect, a pollinator survey transect, and at least half of the co-locations also had a pan-trap station (Fig 2C). By collecting data in this stratified format, comparable community data was collected for each biodiversity indicator within each FC. Furthermore, the data collected at co-locations within defined habitat types in each survey square allowed the relationships between vegetation and pollinator diversity metrics to be assessed, providing further insight. The transects and surveys are described in detail below.

**Breeding bird surveys.** The breeding bird survey used a transect walk method with the option to alternatively conduct point counts if these form part of the country's national breeding bird survey. The notation and monitoring criteria used were based on the protocol outlined in the British Trust for Ornithology's (BTO) breeding bird survey [27]. A single transect line of either 1 km or 500 m (for 500 m x 500 m survey squares) in length was identified in each survey square, running north-south or east-west, in a straight line where possible (Fig 2B and 2C). Transects were ideally 500 m apart (and no closer than 200 m) to avoid double-counts. Depending on transect length, transects were divided into either ten or five 100 m sections and numbered 1–10 or 1–5 respectively. The coordinates of the start and end points of each section were recorded by a handheld GPS device. Monitoring was conducted over two surveys, this is considered sufficient to detect population trends in bird species [28], one between 1st April – 15th May and a second 16th May – 30th June.

Recording birds: Transect lines were walked at a steady pace, with each 100 m section taking around 5 minutes. Birds identified by sight or sound either side of the transect line were recorded using the FRAMEwork Breeding Birds Field Recording Sheet (S3 Appendix). The minimum requirement was to identify the European Farmland Bird Indicator Species [29], though many FCs recorded all species observed. Where possible, species, juveniles, sex, and any breeding behaviours were also recorded (see Breeding Bird Codes and Example Field Sheet in S3 Appendix for details on how to record birds on the recording sheet).

Distance bands: Birds were recorded in one of the following distance bands, according to where they were first noted: 1) within 25 m either side of the transect line; 2) between 25 and 100 m either side of the transect line; 3) >100 m including birds that were outside the 1 km square boundary but within the distance band; and 4) birds that were in flight. Distances were measured at right angles to the transect line. For example, where a bird was seen 100 m ahead within 25 m of the transect line, it was recorded in distance band 1, in the appropriate transect section. Birds performing display flight (e.g., Eurasian skylark, *Alauda arvensis*) or gliding (e.g., common buzzard, *Buteo buteo*) were recorded in the relevant distance band rather than the flyover column and marked with an arrow. Aerial-feeding birds (e.g., barn swallow, *Hirundo rustica*) were recorded as 'in flight', unless they were seen entering a nest site, then they were noted in the appropriate distance category.

 

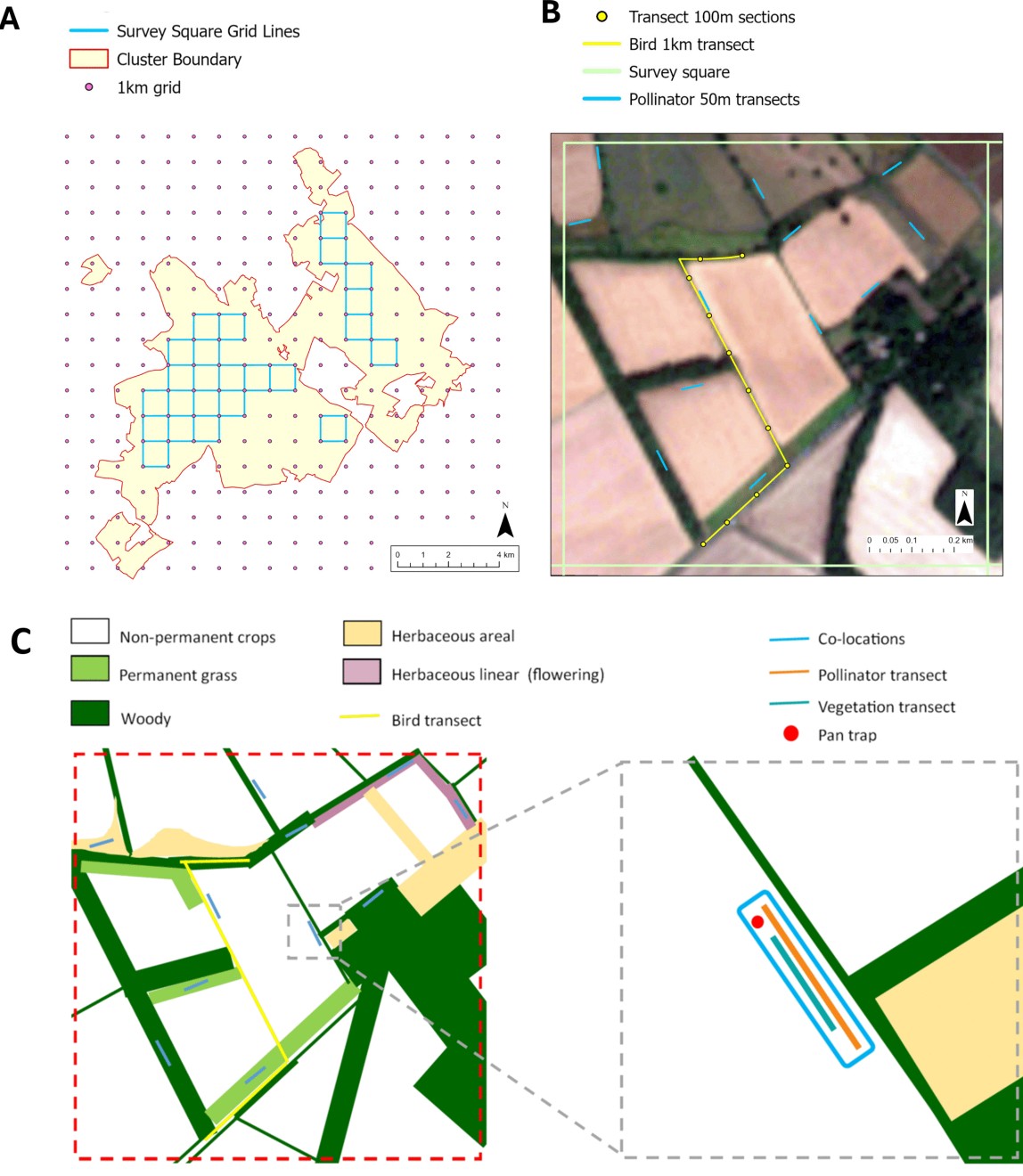

**Fig 2. Farmer Cluster survey squares and transect lines. (A)** The British national grid in pink dots (1 km squares) overlaid on the English Farmer Cluster, with the Farmer Cluster outline in red, and selected survey squares for the English Farmer Cluster in blue. **(B)** A single survey square in the English Farmer Cluster with the breeding bird (yellow) and insect pollinator (blue) transects are illustrated. **(C)** a representation of how the 1 km survey square (in B) might be mapped, with different habitats defined, the 1 km breeding bird survey transects (yellow), all vegetation and pollinator survey co-locations mapped (blue), and the magnified (grey square) showing the layout of the co-locations, each including a 50 m pollinator transects (orange), 20 m vegetation transects (aqua), and with one of the pan trap stations (red).

Survey conditions: Surveys were conducted a minimum of 1 hour after sunrise to avoid bird peak activity and completed by mid-morning. Surveys were conducted in good weather conditions, and weather was recorded using the following BTO codes: cloud coverage – 0–33%, 33–66%, 66–100%; Rain – none, drizzle, showers; wind – calm, light, breezy; and visibility – good, moderate, poor [27].

**Pollinator surveys.** Pollinator surveys were conducted using well-known methodologies already in practice across several countries in Europe which have been proposed for the EU Pollinator Monitoring Scheme (EUPoMS; [30–33]). Butterfly and bumblebee transect walks were combined with pan-traps [30,34–36], to capture maximum insect pollinator diversity of the farms [37,38].

Pollinator transects: *Transect locations.* Surveys were conducted with fixed, short transects of 50 m. The start and end of each transect was clearly marked and recorded with a GPS device (Fig 2B and 2C). These were positioned across each square to provide a representative sample of the different broad habitat classes (BHC) present (Table 2), placed so as to capture the habitat along its length for linear habitats. When areal land parcels were surveyed, the transect was placed >10 m from the boundary to avoid edge effects. The number of transects were adjusted to give approximately 96 transects in total for each FC (S4 Appendix), with equal numbers of transects across the different BHCs.

***Conducting surveys.*** Transects were surveyed a total of four times, once a month between 1st May and 30th August, leaving a minimum of a two-week gap between two successive walks of the same transect [39]. Each transect was walked between 10 a.m. and 5 p.m. on the same part of the day (e.g., morning, early afternoon, late afternoon) within and over the years. Where possible, transects were walked during optimal weather for bumblebees and butterflies, dry with no

**Table 2. Broad habitat classes (BHC).**

| BHC | Type | Description |
|---|---|---|
| Non-permanent crops (including grass in rotation) | Agricultural | All arable and horticultural crops including rotational grass |
| Permanent crops | Agricultural | E.g. soft-fruit, top-fruit, vineyard |
| Permanent grass | Agricultural | Land used to grow grasses or other herbaceous forage naturally (self-seeded) or through cultivation (sown) and that has not been included in the crop rotation of the holding for five years or more (e.g., meadows, pastures, ruderal/ fallow grassland) |
| Herbaceous Areal | Semi-natural | Natural or semi-natural areal elements, e.g., natural grasslands, including abandoned fields with less than 30% shrub/tree canopy cover[b]. Herbaceous areal vegetation can also be sown (flower or grass mixtures) |
| Woody Areal | Semi-natural | Natural or semi-natural areal elements including abandoned fields with more than 30% shrub/tree canopy cover[b] |
| Herbaceous linear[a] (grass) | Landscape element | Any linear structure in which the herbaceous layer is predominantly grass, may also include woody elements (e.g., tree, shrub, or hedge), may be associated with a manmade structure such as wall, fencing, or path, and natural or manmade water courses |
| Herbaceous linear[a] (flowering) | Landscape element | Any linear structure in which the herbaceous layer is predominantly flowering plants, may also include woody elements (e.g., tree, shrub, or hedge), may be associated with a manmade structure such as wall, fencing, or path, and natural or manmade water courses |

Descriptions of each broad habitat class to ensure all habitat types were captured and categorised.

[a]Linear landscape elements are defined as having a width between 2.5 and 5 m, and minimum length of 40 m.

[b]Canopy cover measured as ground projection of the closed canopy layer

 

more than 75% cloud cover, with a minimum temperature of 13°C if the sky was clear (less than 50% cloud), and a minimum temperature of 17°C if the sky was greater than 50% cloud cover [33,40]. It was also ensured surveys were conducted on days where the wind speed was less than 5 on the Beaufort scale.

Each transect was walked at a steady and constant pace, first recording all the bumblebee individuals, and then walked a second time to record butterflies, after a minimum 5-minute gap to allow insects to return. When bumblebee and/or butterfly activity was low, both taxa were counted concurrently. Individuals were recorded (see S5 Appendix for FRAMEwork bumblebee and butterfly transect survey sheet) if seen within a fixed virtual detection box either side of the observer (2 m for bumblebees, 2.5 m for butterflies), and in front of the observer (4 m for bumblebees, 5 m for butterflies). Individuals were identified to (morpho)species level wherever possible in the field (using a net to capture and examine more closely) or were caught and identified in the lab. For each bumblebee observation, the caste (i.e., male, worker, queen) was also noted. Butterfly species were recorded using the updated European named species list [41].

Pan-traps: *Pan trap locations.* Pan traps were deployed in all eleven FCs and were used to target solitary bees and hoverflies that would likely be missed during the transect walks (see S6 Appendix for instructions on making UV pan-traps). Pan traps were positioned at fixed locations across the FC, in areas that provided a representative sample of the different BHC present (Table 2), and were co-located with approximately half of the pollinator transects. Each BHC present in FC's were equally sampled across all survey squares. The number of pan trap locations was adjusted per square to give approximately 48 in total for each FC and control combined (S4 Appendix). Where more than one pan trap was deployed within a given BHC, they were in different parcels or linear elements. The pan traps were also located in areas that would not become heavily shaded, nor would they be disturbed by livestock, members of the public, or farming activity.

When setting up the pan traps, it was ensured that the vegetation surrounding the trap station was not trampled, the height of the pan trap set was equal or slightly higher than the vegetation height, one bowl of each of the three colours (UV blue, UV yellow, and white; [42]) was placed at every trap station, and each bowl filled with ¾ soapy water (Fig 3). The weather conditions and the percentage of bare soil were recorded, as well as XY coordinates and the time, and a photograph of the location was taken. See S7 Appendix for the FRAMEwork pan trap data entry sheet used to record the sampling information.

***Collecting pan trap data.*** Pan traps were deployed once a month in conjunction with butterfly and bumblebee transect surveys, between 1st May and 30th August, four times per surveyed year. Pan traps were set out after the transect walks to avoid interference and capturing of species targeted during the transects. A minimum period of two weeks was maintained between survey visits to any given survey square, and the same weather conditions were met as for the pollinator transect surveys.

Each FC agreed a set length of time that the pan-traps would remain at their locations (between 6–48 hours) based on local prior knowledge of insect attraction rates to pan-traps, and this remained consistent across all trap stations and for all pan trap surveys during the season. Once the allotted time had passed, the pan trap stations were visited in the same order as they were placed. The contents of each bowl were emptied into a tea filter or muslin cloth sat in a tea strainer or sieve (water discarded), and then the filter or cloth was placed directly into a tube with a label detailing the location, date and recorder. The tubes were then placed inside a zip-lock bag and placed in a freezer to store, or a preservation fluid (e.g., 70% ethanol) was added to each tube if a freezer was not available. If ethanol was used, Lepidopteran specimens were removed first and dried in a glassine paper envelope or triangle. If there had been any unusual activity at the pan trap, such as bird faeces or an emptied/overflowing bowl, this was noted on the data collection sheet at the time of collection. Where possible, each bee and butterfly were identified to species (genus or family where species was not possible), hoverflies were counted, and sex of the individual was noted.

***Pan trap floral survey.*** Several studies have shown that the number of pollinators caught in pan traps is often influenced by the number of flowers surrounding a trap [43,44]. Therefore, data on floral resources were collected when the

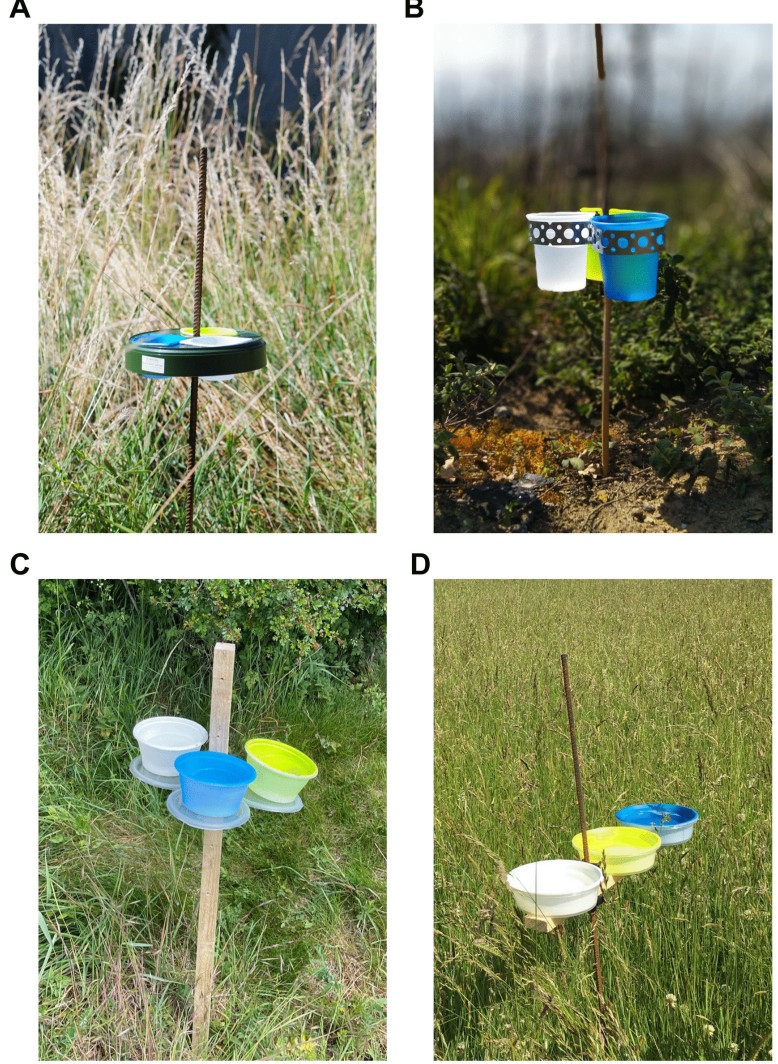

**Fig 3. Examples of pan-trap sets composed of three bowls placed on their support.** Different materials can be used: (A) a plastic tray and an iron rod; (B) metal straps and a cane; (C) plastic lids as trays and a wooden stake; or (D) a wooden support and iron rod. The same pan-trap model must be used consistently within each Farmer Cluster. Bowls should be positioned at equal or slightly above the average vegetation height.

trap was set or recovered. On either the setting or collection day, the flowers present were counted within a 2 m radius of each pan trap station [32]. All fresh herbaceous dicotyledonous plants (i.e., no grasses) in bloom were recorded to species level (photographs were taken for expert identification where necessary), and all 'flower units' within the 2 m radius counted. A 'flower unit' (S8 Appendix) was defined as either a single flower or an umbel, spike or capitulum on a multi-flowered stem [32,45]. Additionally, the average height of most abundant vegetation was also recorded.

**Vegetation and habitat survey.** Vegetation and habitat surveys were conducted to estimate the vegetation structure, the plant diversity and the nature value (i.e., biodiversity value) of the FC landscape. The protocols were based on similar projects whose methods were shown to be effective and could be easily implemented across all FCs [19,24,46].

The vegetation plots: Vegetation plots were all co-located with each of the pollinator transects at fixed locations within the survey squares, providing a representative sample of the habitats present within the squares (BHCs, Table 2). Each

vegetation plot was 20 x 2.5 m in size (except for two FCs that chose to extend the plots to 50 m), ran parallel to the pollinator transect, covered one BHC, and was positioned to cover the primary herbaceous vegetation along that transect. The start and end of each plot was clearly marked and recorded with a GPS device.

Conducting the surveys: Vegetation surveys were carried out 2 times per year during a short time frame (i.e., all plots were surveyed within a 2–3 week window): one in late spring when the vegetation was at the peak of the growing period and well developed, with many flowering species, but before the first cut or the start of the grazing period if possible (i.e., mid-April to late-June); and the second later in the summer (from early-July to late-August; though one FC survey ran into early September). Dates were adapted to accommodate the farming practices and local climatic conditions for each FC.

Observations were made whilst walking the length of the plot, and the following parameters (Table 3) were recorded (see S9 Appendix for survey sheets).

## Anticipated results

The above-described biodiversity monitoring protocols allowed for the collection of a large amount of quantitative data, working within a broad range of farming systems, landscapes, and habitat types across Europe. During the baseline year, the stratified sampling method (survey squares) captured both the habitats (BHCs) and biodiversity across 103.75 km$^2$ of land. The surveys included 124 km of bird transects, 19.35 km of vegetation transects, 55.7 km of pollinator transects, 266 pan trap stations were deployed for 557 hours, overall resulting in more than 3700 person hours in the field and more than 2600 person hours conducting additional species identification in the laboratory (Table 4). Using both internal project partner experts and contracted external experts to conduct all the fieldwork and species identification, the baseline surveys recorded over 2000 individuals (Table 4). The use of aligning biodiversity indicator transects (co-locations) will allow the data to be amalgamated within each survey square, bringing together the biodiversity indicators for further comparative analysis (Fig 4).

The data collected will allow multiple avenues of analysis to be considered when addressing the impact FCs have on biodiversity, a non-exhaustive list of examples is provided here. In terms of the BACI analysis, if the null hypothesis is rejected, the FCs will show an increase in biodiversity over time in comparison to the control sites (illustrated by Fig 4A). Landscape descriptors (e.g., SNH coverage, habitat fragmentation) should also be statistically compared between the FC and control squares to adjust for any confounding impacts. Conducting analysis for each biodiversity indicator and each FC separately allows identification of which indicators and FCs performed the best. This opens a discussion of why certain FCs or indicators have seen improvements, and whether the FC approach could be considered for improving overall farmland biodiversity across Europe. Additionally, the vegetation diversity metric and its relationship with the pollinator diversity metric will likely vary between habitat types (BHC) and can be assessed through a linear relationship for each FC. For example, floral vegetation may have the highest vegetation diversity metric and the strongest positive relationship with the pollinator diversity metric (Fig 4B). This will allow for a discussion of which habitats and biodiversity sensitive farming actions might have the most positive impact on vegetation and pollinator biodiversity (e.g., reduced pesticide use, sowing of wildflower strips), and how this can impact overall farmland biodiversity. Furthermore, community dissimilarity analysis will determine the differences in species communities between the baseline year (before) and final year (after) of individual FCs (Fig 4C). If the null hypothesis is rejected and there is an increase of biodiversity in the FCs, the community of species found would change over time, and the 'after' community would be strongly associated with biodiversity (Fig 4C). Finally, a before-after comparison of different biodiversity metrics in each FC will show which FCs saw the greatest improvements (Fig 4D). This allows discussion of why a certain FC might have seen these improvements.

By linking the biodiversity outcomes with other factors such as farmer engagement, financial incentives [47], and management changes [48], this could provide a more in-depth understanding of how the FC is working and which socio-economic indicators are linked to improved farmland biodiversity across FCs.

**Table 3. Parameters to be recorded in the vegetation plots.**

| Parameters | Categories or attributes of parameters |
|---|---|
| **For all habitats** | |
| Plot ID | Survey square, treatment (Farmer Cluster or control), BHC, co-location ID, transect ID, aligning pollinator transect ID |
| Date, observer, GPS coordinates | Date of survey; name of person conducting survey; GPS coordinates of start and end point of plot |
| Land cover classification (LC1/2) | Code of the land cover and type of culture for crops[a]<br>LC1 is the ground layer: arable land, grassland, fallow land; and LC2 is typically the woody layer: could be a permanent culture such as fruit trees, vineyards or berries. |
| Wood/Tree/Bush elements | 3-point nature value score: 0: No wood/tree/bush elements; 1: Small and/or young trees, or small, closely cut hedges with few woody species and low structural richness; 2: Trees of medium size and age, hedges or scrub of moderate size and density including >2 species; 3: Large and/or old trees with obvious habitat features such as hollows or dead branches. Hedges or scrub are richly structured with layers at different heights, and include several different woody species |
| 1. Transect info | Has the transect shifted – if yes, why, and new GPS coordinates |
| Stage of the land | Visible – not yet visible – mowed/harvested |
| 2. Photo ID of the plot | Taken from the start, the end, the flower posy and vegetation structure |
| 3. Site exposition | Direction of slope: N, W, S, E, NE, SE, SW, NW |
| Inclination of slope | Flat = (0) = 0%; (1) 0–3%; (2) 4–8%; (3) 9–15%; (4) 16–25%, (5) 26–40%, very steep slope (6) ≥40% |
| **For arable land** | |
| 4a. Coverage of crop, wild plants | The coverage is divided into the ground layer (<1.5 m) and the canopy layer (>1.5 m). The ground layer may consist of different components, but the coverage of these must sum up to 100%. Crop: The coverage of the crop plants is estimated on the transect area in %. If a field has been sown but the seedlings are not yet (or only just) visible, 1% should be selected. Wild plants: The coverage of non-crop (wild) plants is estimated in % on the transect area, excluding crop species that are growing spontaneously from the previous crop ("volunteers"). Underseed: Some annual crops are undersown with a seed mix to suppress weeds, for example cereals undersown with grass. Underseed plants show a much more regular pattern than wild plants. Bare soil and stones: Open soil with bare ground or stones which can be seen from above by looking down at the ground from approximately 1.5 m height (a standing position). Woody layer on the ground: these are small or dwarf shrubs <1.5 m in height. Woody plants in the ground layer are very rare in arable fields.<br>The canopy layer is all woody plant material >1.5 m in height and recorded in % in addition to the ground layer. Orchard: usually fruit trees that shade a crop. Olive groves: olive production can be mixed with arable crops. Others: all other types of trees. |
| 4b. Height of crop | Average height of the crop in cm. |
| 5a. Total number of flowering species | Number of species counted – count the total number of currently flowering forb species (also the species which have well developed flower buds almost ready to flower, and species which have just finished flowering and the dead flowerheads can be seen). Remember to check for very small flower species (e.g., *Stellaria media*). |
| 5b. Flower density | On a scale of 0–5, where 0 is very sparse, and 5 is very dense. |
| 6. Flowering colours | Which flower colours are present. |
| 7. Record of indicator species/ species groups | Presence of indicator species – while walking slowly along the transect line, the indicator species that are present should be noted, presence only – no estimation of coverage. Lists of the indicator species for grassland and arable systems are provided (see S7 Appendix). |
| Record of all the species with cover (optional) | All plant species and cover. Each species is associated to the cover recorded in the plot using 5% categories. |
| 8. Remarks | Any comments on plot |

*(Continued)*

**Table 3.** (Continued)

| Parameters | Categories or attributes of parameters |
|---|---|
| ***For grassland*** | |
| 4. Vigour of the vegetation | Grassland types are often characterized by their often quite different types of vegetation height and density (vigour). Vigour is indicated when the vegetation is in full development in spring; thus grazed plots may be judged with the help of ungrazed patches remaining on the pasture: very meagre/sparse, meagre growth, light growth, dense growth, very dense/mass growth. |
| Graminoid-forb ratio | Enter values for the ratio of cover of graminoids (grasses, sedges, rushes, reeds) to forbs (i.e., non-woody, broadleaved plants including fern species). The sum is always 100, also if the total coverage of this layer is only 50% (and 50% of bare layer or 50% of bushes and dwarf shrub). Thus, the values can be indicated as follows: 100:0, 90:10, 80:20, 70:30, 60:40, 50:50, 40:60, 30:70, 20:80, 10:90, or 0:100. |
| 5a. Coverage of vegetation layers | Graminoid-herb layer, shrub layer, trees layer, litter moss or lichen, bare soil as percentages. Open soil with bare ground or stones which can be seen from above by looking down at the ground from approximately 1.5 m height (a standing position). Woody layer on the ground: these are small or dwarf shrubs <1.5 m in height. Woody plants in the ground layer are very rare in arable fields.<br>The canopy layer is all woody plant material >1.5 m in height and recorded in % in addition to the ground layer. Orchard: usually fruit trees that shade a crop. Olive groves: olive production can be mixed with arable crops. Others: all other types of trees. |
| 5b. Height of the herbaceous layer | Average height of the herbaceous layer in cm. |
| 6. Grassland habitat type | Meadows, Pastures with scale of management intensity or other grassland types. |
| 7a. Grassland age | Grassland older than 5 years (> 5 years): species-rich composition (normally more than 10 herb and grass species present), and no seeding rows from a seeding drill visible, the sward is rather dense and the composition/ distribution of the grass and herb species on the ground is balanced. (Please note: very intensively managed meadows, with high slurry inputs after each of several cuts per year show less than 10 plant species and still are old in the sense of same use and no ploughing for many years.)<br>Grassland 5 years old or younger (≤ 5 years): seeding rows of grassland are visible and/or only few grass and forb species occur (< 10 species), patches of open ground<br>Unclear: If there are no clear signs, choose this option |
| 7b. Grassland fertilisation | No: no additional fertiliser has been applied<br>Probably: the grass is lush and dark green and seems to be fertilised regularly but there are no direct signs of fertilisation; sward consists only of few species<br>Sure: direct signs of fertilisation (e.g., slurry, manure or mineral fertiliser on the surface)<br>Unclear: there are neither positive signs nor negative signs of grassland fertilisation |
| 8a. Number of flowering forbs | Number of species counted – count the total number of currently flowering forb species (also the species which have well developed flower buds almost ready to flower, and species which have just finished flowering and the dead flowerheads can be seen). Remember to check for very small flower species (e.g., *Stellaria media*). |
| 8b. Flower density | On a scale of 0–5, where 0 is very sparse, and 5 is very dense. |
| 9. Flowering colours | Which flower colours are present. |
| 10a and 10b. Presence and cover key species and structural characteristics | Certain species tend to dominate under certain conditions. These may indicate positive conditions for grassland nature value (signs of extensive use) or negative conditions (e.g., abandonment).<br>If the species on the list in the form are present on the transect, they should be selected and their coverage in % recorded. |
| 10c. Total cover of legumes | Cover in % of all legume species. |
| 11. Remarks | Any comments on the plot |

The EMBAL survey [19] parameters were followed closely, adding in necessary factors to fit the FRAMEwork methods.

[a]See Oppermann *et al*. [19] for full list of land covers.

**Table 4. Farmer Cluster biodiversity surveys.**

| Farmer Cluster details | | | | Breeding bird surveys | | | | | Pollinator surveys | |
|---|---|---|---|---|---|---|---|---|---|---|
| Location | No. of cluster squares | No. of control squares | Size of squares (km2) | Survey type | No. transects | Length of tran-sects (km) | Survey/ID conducted by | Total individuals counted | No. transects | Length of pollinator transects (m) |
| Austria, Burgenland | 8 | 8 | 0.25 | Transect | 16 | 0.5 | Internal Expert | 133 | 96 | 50 |
| Austria, Mostviertel | 19 | 4 | 0.25 | Transect | 15 | 0.5 | Internal Expert | 232 | 160 | 50 |
| Czech Republic, Velké Hostèrádky | 4 | 4 | 1 | Transect | 8 | 1 | External Expert | 156 | 96 | 50 |
| England, Cranborne Chase | 24 | NA | 1 | Transect | 24 | 1 | Internal Expert | 395 | 115 | 50 |
| Estonia, Southern Estonia | 7 | 6 | 1 | Transect | (26) | (1) | (External Expert) | 197 | 24 | 50 |
| France, Basse-Durance Valley | 12 | 5 | 0.25 | Transect | 18 | 0.5 | External Expert | 74 | 170 | 50 |
| Italy, Calci | 6 | 6 | 1 | Transect | 12 | 1 | External Expert | 123 | 144 | 50 |
| Luxembourg, Born | 4 | 4 | 0.25 | Transect | 8 | 0.5 | Facilitator and internal expert | 178 | 96 | 50 |
| Netherlands, Zeeas-terweg, Lelystad | 8 | 4 | 1 | Transect & point observa-tions | 10 | 1 | External Expert | 85 | 37 | 50 |
| Scotland, North East Scotland | 5 | 5 | 1 | Transect | 10 | 1 | External Expert | 272 | 96 | 50 |
| Spain, Aguilar de la Frontera | 5 | 4 | 1 | Transect | 9 | 1 | Facilitator | 120 | 90 | 50 |

Details of all baseline biodiversity surveys completed in each Farmer Cluster and control, including the Farmer Cluster survey square details, number of transects, length of transects, surveyor required, and number of person hours to conduct surveys. Data entered in brackets refers to surveys completed in 2021 and retained as the baseline for the project, all other baseline surveys done in 2022.

## Discussion

Evidence continues to highlight the significance of considering farmland biodiversity at the landscape scale [49] and the importance of managing landscapes collectively to achieve meaningful outcomes [50]. FRAMEwork brings together the expertise of multiple institutes to address the ongoing decline in farmland biodiversity. By working through FCs and promoting a bottom-up approach to biodiversity sensitive farming, the project aims to develop a method for improving and monitoring biodiversity appropriate for a range of European landscapes.

To this end, FRAMEwork has adapted and tested well established biodiversity monitoring protocols, creating a novel, comprehensive and replicable process that aims to track biodiversity changes across a variety of farming systems, specifically at a multi-farm scale. The protocols presented here provide a practitioner-focused tool that can generate consistent and comparable biodiversity data in new and established FCs. They have undergone multiple rounds of refinement to not only improve the detail of the protocol, but to make them fully applicable in a range of countries, landscape types and farming systems across Europe. Here we evaluate our methods.

| Pan traps | | | | Vegetation surveys | | | | | | Effort | |
|---|---|---|---|---|---|---|---|---|---|---|---|
| Survey/ID conducted by | Total individuals counted | No. pan trap locations | Length of deployment (h) | Survey/ID conducted by | Total individuals counted | No. plots | Size of plot (m X m) | Survey/ID conducted by | Total plant genera counted | Total no. person hours (surveys) | Total no. person hours (additional species ID) |
| External Expert | 68 | 48 | 20 | External Expert | 94 | 96 | 20.0 x 2.5 | Internal Expert & facilitator | 12 | 600 | 350 |
| External Expert | 119 | 48 | 24 | External Expert | 35 | 91 | 20.0 x 2.5 | Facilitator | 9 | 360 | 200 |
| Internal Expert | 99 | 48 | 6 | Internal Expert | 38 | 96 | 20 x2.5 | Internal Expert | 18 | 480 | 200 |
| Internal Expert | 66 | (36) | (48) | (Internal Expert) | 26 | 115 | 20.0 x 2.5 | External Expert | 10 | 380(176) | (185) |
| Internal Expert | 57 | 48 | 24 | External Expert | 50 | 48 | 50.0 x 2.0 | Facilitator | 12 | 375 | 700 |
| Internal experts and Facilitator | 78 | 51 | 24 | N/A | 41 | 51 | 20 x 2.5 | Facilitator, internal and external experts | 14 | 630 | N/A |
| External Expert | 53 | 144 | 24 | Internal Expert | 42 | 72 | 20 x 2.5 | Internal Expert | 25 | 800 | 200 |
| Facilitator | 81 | 48 | 24 | Facilitator | 54 | 96 | 20 x2.5 | Facilitator | 17 | 278 | 250 |
| Internal Experts | 97 | 37 | 48 | Internal Expert | 33 | 37 | 20 x 2.5 | Internal Experts | 9 | 200 | 112 |
| Internal Expert | 75 | 48 | 48 | Internal Expert | 58 | 96 | 20.0 x 2.5 | Internal Expert | 14 | NA | NA |
| Facilitator | 186 | 40 | 24 | External Expert | 32 | 45 | 20 x 2.5 | Facilitator | 21 | 184 | 750 |

## Farmland birds

Farmland birds include primary and secondary consumers and as such provide a useful biodiversity indicator responding to lower trophic levels on which they depend for food (seeds, insects etc.) and nesting resources (habitat availability) [51]. The European Farmland Bird Indicator (EFBI) was created by Gregory et al. [51], and since then it has been used to predict that multiple European countries would witness a decline in farmland bird species in response to agricultural land-use change [52], making farmland birds a significant landscape-scale biodiversity indicator to monitor, hence their inclusion in many similar projects [16,24,25]. However, it required a high level of expertise to identify all potential farmland bird species by sight and sound, and the BBS annotations require practice, meaning skilled individuals were needed to conduct the surveys [25]. Alternatively, either using point-count locations along the transect and a bird species identification application (e.g., Merlin Bird ID App) could greatly reduce the expertise level required, or the discovery of country- and system-specific indicator species that signify a diverse farmland bird community could encourage a wider range of practitioners to become involved in the surveys [53]. Additionally, some species that are often considered key indicator species might be under-recorded or missed with their monitoring requiring the use of species-specific survey methods (e.g., grey partridge; [54]. Because the FRAMEwork bird surveys were aligned to use the annotations from the BBS survey method, a transect line walk was opted for. Transect walks suit open habitats

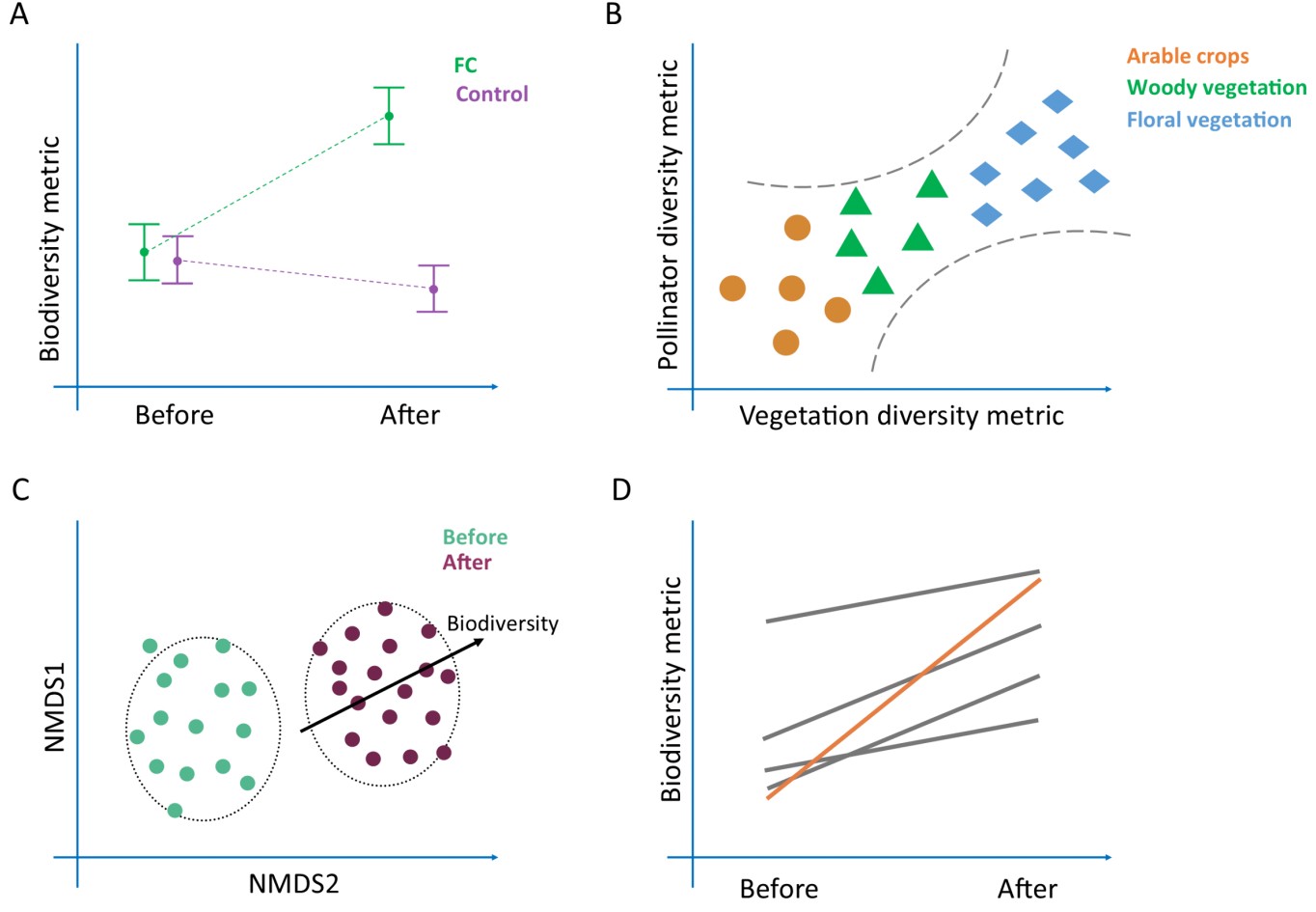

**Fig 4. Hypothetical measures of the impact of Farmer Clusters on biodiversity. (A)** Results of the before-after-control-impact analysis comparing the Farmer Cluster to its control for a biodiversity metric. **(B)** Relationship between the vegetation and pollinator biodiversity metrics for co-locations within different broad habitat classes. **(C)** Community analysis of a Farmer Cluster comparing the baseline year (before) and the final year (after) surveys, where an increase in biodiversity is significantly associated with the 'after' community. **(D)** A comparison of the different Farmer Clusters showing improvements in a biodiversity metric, with the Farmer Cluster showing the most significant improvement over time highlighted (orange).

such as farmland, allowing many individuals to be identified efficiently along the walk, with double counts and errors in distance estimates having a minor impact on the data collected [55]. Similar transect line surveys are already conducted as part of national bird census initiatives across Europe [56–58], with the data often used in large-scale biodiversity monitoring projects [20] and collated through the Pan-European Common Bird Monitoring Scheme, hence this method was chosen to be implemented in the FCs. However, some landscape-scale projects opt for different methods, such as "area searching" (mapping all birds that were seen or heard; [25]) or point counts, the latter with the potential to be used in place of transect walk. For example, the Zeeasterweg, Lelystad FC in the Netherlands conducted a point count for each transect/survey square to align more closely with their country's national monitoring scheme [59]. Additionally, the Bass-Durance FC in France began their breeding bird surveys earlier than the protocol specified to capture migratory species as per their national bird survey, illustrating the flexibility of the overall approach and its ability to be adapted to satisfy local requirements. Future farmland bird surveys could be adjusted as described here to allow flexibility around the FC's local conditions.

## Insect pollinators

Insect pollinators were an important group to consider, used as indicators in many similar projects [16,20,24,25] as they not only add to the overall biodiversity of the landscape but provide a vital pollination service to many crops and wild plants [60–62] and occupy a low trophic level in food webs. Although different species vary in their mobility (e.g., [63–70]), they demonstrate high levels of sensitivity to environmental changes at the landscape-scale, particularly to floral richness, land use, and grassland management [63,71–74]. Among insect pollinators, butterflies are one of the best-known groups and are monitored across Europe by the European Butterfly Monitoring Scheme (eBMS; [75]). They are used in multi-species European indicators, such as the European Grassland Butterfly Indicator [76]. Bumblebees are also monitored in various countries, for example, in the UK by the Bumblebee Conservation Trust BeeWalk scheme [77], or in Luxembourg by the Luxembourg Pollinator Monitoring Scheme [78], but have not yet been included in European indicators. However, these wild bees are among the most important pollinators of many wild, agricultural, and horticultural plants, including fodder and oilseed plants as well as many fruits [79]. On one hand, as generalist foragers, changes in their populations can significantly affect plant and associated animal communities [80,81]. On the other hand, the persistence of their colonies requires the availability of sufficient and diverse floral resources within their foraging ranges (i.e., from less than 1 km for smaller species to more than 3 km for larger species; [64–68]. Other wild bees, such as solitary bees, can also provide a significant pollination service to crops [82] and wildflowers [61,83] due to the method of loose pollen transportation on the hairs of their body [84]. Although they typically have shorter flight distances than bumblebees [69,70], their diversity is often linked to the presence of key plant species in the vicinity [85,86]. Therefore, wild bees are highly susceptible to a decrease in habitat diversity and fragmentation within the landscape [87,88], making them a relevant biodiversity indicator at the landscape-scale.

Evidence on the importance of considering non-bee insect pollinators is increasing due to the large number of flower visits they make in comparison to bee visits [89]. Adult hoverflies are considered generalist foragers, with species typically visiting a wide range of flower species [90] (though see [91,92] for evidence of specialisation). Furthermore, the larvae of certain species predate on aphids, and can contribute to natural pest control as part of an Integrated Pest Management strategy [93,94]. Therefore, the inclusion of hoverflies as a biodiversity indicator allows for the consideration of biological control in the landscape. In addition to butterflies, both wild bees and hoverflies are now explicitly recognised as key indicator groups in the EU-wide biodiversity monitoring framework mandated under Article 10 of the Nature Restoration Regulation (EU) 2024/1991 [95]. Their inclusion in the EU Pollinator Monitoring Scheme (EUPoMS; [31]) is further detailed in the Delegated Regulation (EU) 2025/2188 [96]. These developments underscore the strategic importance of incorporating wild bees and hoverflies in FRAMEwork in alignment with current EU biodiversity and restoration policy.

Transect walks and pan-traps were chosen for the pollinator surveys as complementary methods. Transect walks allow larger individuals such as bumblebees and butterflies to be easily detected, and are widely used in citizen science monitoring programs to assess pollinator populations (e.g., [33,77,78]. Pan-traps, however, capture smaller individuals that would be missed during a transect walk [38]. Additionally, transects often reflect local floral resources, whereas pan-traps are influenced by the surrounding landscape context [97], highlighting the value of combining both methods in landscape-scale monitoring such as that undertaken by FCs.

Although butterfly transect walks are typically about 1 km in length [80,98], the FRAMEwork pollinator protocol originally trialled a 3 km continuous transect to capture a representation of the pollinators in the survey squares, similar in length to other projects that used longer transects [20,24,25]. However, this method resulted in over-representation of crops in survey squares/countries with larger farms and farm fields. Instead, the protocol was modified to shorter multiple 50 m transects in a stratified approach where transects were placed in distinct habitats in numbers representative of the survey square. This ensured no habitat type was over-represented and would work across all European Farmer Cluster landscapes and farming systems.

Those conducting the transect walks had to be confident in identifying multiple pollinating insect species 'on the wing', capturing specimens in a net for closer inspection in the field, or euthanizing specimens to take with them for identification under a microscope (with additional specimens from the pan-traps). Future FCs would therefore require a facilitator skilled in this biodiversity indicator or otherwise have to incur costs of external contracting. There are a few non-lethal technologies available (e.g., the DIOPSIS camera trap; [99]), but most are emerging or highly expensive, which would make them impractical to implement in all the FCs of the project or expect future FCs to finance them. To fully understand the insect pollinator community and its potential benefits to overall biodiversity, its relationship to the surrounding landscape, and any ecosystem services provided, species-level identification is preferable. However, this was beyond the capacity of many project partners, and abundance counts at group levels were therefore considered sufficient. Furthermore, a recent large-scale synthesis across European habitats [97] found that total wild bee abundance was a very strong predictor of wild bee species diversity across different habitats and sampling methods, while bumblebee abundance alone was a weaker proxy. These findings indicate that incorporating total wild bee abundance into our surveys, alongside bumblebee counts, could improve the robustness of our transect protocol and provide a more reliable proxy for community-level pollinator diversity. Therefore, if identifying insect pollinators to species level is outside the abilities or funding of a future biodiversity survey, collecting wild bee abundance data could be used as an alternative proxy.

## Vegetation

As primary producers and the start of many food chains, high levels of plant diversity and abundance are vital for the overall health of an agroecosystem. The richness of vascular plants in a habitat has been correlated with overall levels of biodiversity found [100], and the presence of indicator plant species are commonly used to describe the biodiversity of a habitat or landscape [71]. In addition to noting plant species and flower colour groups present, recording the habitat types, structures, and canopy layers allows a better overall assessment of the habitat diversity within the landscape [19]. Changes in vegetation (species, features, and structures) are often the first signs of a shift in habitat management [101], making plants a key indicator to monitor changes in farming practices and the associated levels of biodiversity in the FRAMEwork project. Recording plant structure and percentage cover is a common theme in large-scale biodiversity monitoring projects [19,20,24,25], and so was considered a highly important biodiversity indicator. Although plants could be deemed as static and not directly representative of movement across the landscape, their pollen and seeds can be dispersed over varying distances by wind and/or animals (pollinators in the case of pollen, and birds or mammals for seeds; [102,103], making them still relevant to consider as a landscape-scale biodiversity indicator. The FRAMEwork vegetation surveys were primarily based on the EMBAL survey [19] using a transect walk (e.g., [25]), which as a European protocol should be readily recognisable by multiple project partners and future FCs across Europe. Although plant surveys of some similar projects are conducted in plots or quadrats [24,104], to better capture the vegetation within the pollinator transect, a 20m length transect was opted for. Due to increasing numbers of plant identification applications and artificial intelligence tools (e.g., PlantSnap Inc), most plants can be easily identified in the field at the time of the survey, making vegetation surveys accessible to a greater number of practitioners. However, there were lots of components to the survey making it complex and time consuming, and future FCs could consider switching to a single survey conducted at peak flowering season (May-June depending on climate and local conditions, recommended in the EMBAL guidelines [19]) instead of two surveys to relieve the survey effort required. Additionally, the surveys didn't capture the open flowers present specifically during the pollinator surveys, resulting in only inferred relationships between vegetation and insect pollinators recorded.

## Additional biodiversity monitoring

Although other environmental indicators, services, and pressures were considered and are ecologically important, the selected taxa reflected both the priorities of the FRAMEwork project and the need to ensure feasibility within available

resources. For example, moths (macro and micro) and bats were of interest to a number of project partners, but the inclusion of these groups would have significantly increased the required species identification skills and person-hours. However, with the ongoing advancement of technologies and automated identification tools (e.g., those being developed through the eBMS [105]), monitoring these groups may become more feasible in future FCs. Decisions to expand monitoring to additional taxa should carefully consider the aims of each FC, funding availability and technical support. While the current suite of biodiversity indicators described above are recommended as a practical and ecologically meaningful starting point, the inclusion of additional species groups is encouraged where relevant and feasible. Additionally, environmental pressures such as pesticide use, fertiliser applications, water quality, and area of semi-natural habitat are all important indicators that were considered through a separate FC sustainability analysis [48], and ecosystem service indicators along with habitat suitability metrics were considered through the Farmland Ecosystem Assessment Support Tool (FEAST; [106]). These could be combined alongside the biodiversity indicators suggested above, should the FC have the funding and resources to include them.

The FRAMEwork survey square and transect placement protocol provides a template to monitor additional biodiversity indicators. For example, pitfall traps (e.g., for beetles), light traps (e.g., for moths), or ink footprint tunnels (e.g., for small mammals/hedgehogs; [107,108]) at singular points within each habitat type (mimicking the pan trap layout), could be deployed. Alternatively, sweep nets along the transects already identified (e.g., for foliar insects; [109]); or camera traps (e.g., for mammals) and acoustic recording devices (e.g., for birds or bats; [110]) could be placed one per square or one per habitat type within each square.

Furthermore, there is a gradual shift towards AES payments becoming 'results-based' (RPBS; [71]). These schemes often require collaboration across the landscape to see true species recovery [111]. Not only do FCs provide collaboration at the landscape-scale of the FC, but the above-described additional biodiversity indicators show the ease at which a targeted RBPS species could be integrated into the biodiversity monitoring template. This would result in standardised monitoring of the target species, ensuring that the biodiversity levels required to meet the expectations for payment are recorded.

## Conclusions

Farmer Clusters bring together neighbouring farmers to address biodiversity restoration at the landscape-scale, creating a sense of community and shared investment in the landscape. They provide an opportunity to test a "bottom-up" approach to farmland conservation, each farmer implementing tailored and targeted biodiversity-friendly actions on their land in collaboration with the rest of the FC. However, to understand how biodiversity is changing in the FCs, standardised biodiversity monitoring is required. Provided here is a detailed protocol, describing how to stratify the sampling across the FC landscape, select appropriate biodiversity indicators, survey square sizes, and transect lengths, and the biodiversity survey methods for six landscape-scale biodiversity indicators (including survey sheets).

This paper assesses the validity of the methodology, providing sound evidence for the inclusion of each biodiversity indicator, and the survey methods used to monitor changes in each. This paper can be used by practitioners as a tool to monitor biodiversity across the landscape of any current or future FC, with the suggestion that surveys are conducted at the inception of the FC and then very 3–5 years to capture the long-term changes across the FC.

## Supporting information

**S1 Appendix. BACI calculation equation.**
(DOCX)

**S2 Appendix. All biodiversity indicators and difficulty calculation method.**
(DOCX)

**S3 Appendix. FRAMEwork Farmland bird survey sheet and notes.**
(DOCX)

**S4 Appendix. Transect spread.**
(DOCX)

**S5 Appendix. FRAMEwork pollinator transect survey sheet.**
(DOCX)

**S6 Appendix. Pan trap details.**
(DOCX)

**S7 Appendix. FRAMEwork pan trap survey sheet.**
(DOCX)

**S8 Appendix. Flower unit and plant indicator species.**
(DOCX)

**S9 Appendix. FRAMEwork vegetation survey sheet.**
(DOCX)

## Acknowledgments

We would like to express our gratitude to all members of the FRAMEwork project whose expertise and insights were invaluable to the development of this project. We also acknowledge the time given by project facilitators in the testing of these methods during the trial year. And finally, to all the farmers and land managers for collaborating with us for the course of the project.

## Author contributions

**Conceptualization:** Graham S. Begg, Lisette Cantú-Salazar, John M. Holland, John Tzilivakis, Douglas J. Warner, Niamh M. McHugh.

**Funding acquisition:** Graham S. Begg, Lisette Cantú-Salazar, John M. Holland, John Tzilivakis, Douglas J. Warner, Niamh M. McHugh.

**Methodology:** Graham S. Begg, Lisette Cantú-Salazar, John M. Holland, Youri Martin, John Tzilivakis, Sarah Vray, Douglas J. Warner, Niamh M. McHugh.

**Project administration:** Graham S. Begg.

**Visualization:** Rachel N. Nichols, Alon Zuta.

**Writing – original draft:** Rachel N. Nichols.

**Writing – review & editing:** Rachel N. Nichols, Graham S. Begg, Lisette Cantú-Salazar, John M. Holland, Youri Martin, John Tzilivakis, Sarah Vray, Douglas J. Warner, Niamh M. McHugh.

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
