## [Decision Letter · Decision Letter 0]

25 Dec 2025

Dear Dr. Nichols,

Thank you for submitting your manuscript to PLOS ONE. After careful consideration, we feel that it has merit but does not fully meet PLOS ONE’s publication criteria as it currently stands. Therefore, we invite you to submit a revised version of the manuscript that addresses the points raised during the review process.

**Please revised the manuscript according to the comments of the reviewers.**

We look forward to receiving your revised manuscript.

Kind regards,

RunGuo Zang

Academic Editor

PLOS One

Journal Requirements:

“This Project has received funding from the European Union's Horizon 2020 research and innovation programme under grant agreement No 862731”

3. We note that you have referenced Nichols et al.” and “McHugh et al.” which has currently not yet been accepted for publication. Please remove this from your References and amend this to state in the body of your manuscript: “Nichols et al., unpublished” and “McHugh et al., unpublished” as detailed online in our guide for authors

4. We note that Figures 1 and 2 in your submission contain map and satellite images which may be copyrighted. All PLOS content is published under the Creative Commons Attribution License (CC BY 4.0), which means that the manuscript, images, and Supporting Information files will be freely available online, and any third party is permitted to access, download, copy, distribute, and use these materials in any way, even commercially, with proper attribution.

For these reasons, we cannot publish previously copyrighted maps or satellite images created using proprietary data, such as Google software (Google Maps, Street View, and Earth). For more information, see our copyright guidelines: http://journals.plos.org/plosone/s/licenses-and-copyright.

1. You may seek permission from the original copyright holder of Figure(s) [#] to publish the content specifically under the CC BY 4.0 license.

5. We note that Figures 3 and s10 in your submission contain copyrighted images. All PLOS content is published under the Creative Commons Attribution License (CC BY 4.0), which means that the manuscript, images, and Supporting Information files will be freely available online, and any third party is permitted to access, download, copy, distribute, and use these materials in any way, even commercially, with proper attribution. For more information, see our copyright guidelines: http://journals.plos.org/plosone/s/licenses-and-copyright.

1. You may seek permission from the original copyright holder of Figure(s) [#] to publish the content specifically under the CC BY 4.0 license.

Additional Editor Comments:

Please respond to the concerns of the referees and revised the manuscript accordingly

Reviewers' comments:

Reviewer's Responses to Questions

**Comments to the Author**

1. Is the manuscript technically sound, and do the data support the conclusions?

Reviewer #1: Yes

Reviewer #2: Partly

2. Has the statistical analysis been performed appropriately and rigorously?

Reviewer #1: N/A

Reviewer #2: I Don't Know

3. Have the authors made all data underlying the findings in their manuscript fully available?

Reviewer #1: Yes

Reviewer #2: No

4. Is the manuscript presented in an intelligible fashion and written in standard English?

Reviewer #1: Yes

Reviewer #2: Yes

Reviewer #1: Comments on MS PONE-D-25-36303 entitled “A standardised protocol for measuring farmland biodiversity outcomes across European Farmer Cluster landscapes” by Nichols et al.

1. General comments:

This paper proposes a standardized protocol for monitoring farmland biodiversity in European Farmer Cluster landscapes. The protocol is designed to evaluate and unify methods for assessing biodiversity across different farming systems. It provides practical guidance on selecting survey squares and transects, as well as detailed procedures for surveying birds, pollinators, and vegetation. Comprising a suite of standardized sub-protocols, the framework is adaptable to diverse agricultural contexts. Finally, the authors evaluate the effectiveness of these biodiversity indicators and survey methods. This paper is of good scientific quality, thus may interest the readers of PLoS One. However, there are some aspects that could be improved for me. Therefore, I recommend this paper to have a major revision. After this major revision, the paper may contribute to the field.

2. Specific comments

1) Is it necessary to include some indicators of biodiversity pressures such as the annual application of pesticides, annual application of fertilizers, per capita annual net income of rural households and so on. These indicators could facilitate the subsequent analysis after biodiversity investigation.

2) Is it necessary to include some indicators of ecosystem services such as water purification, reducing pesticide needs and so on.

3) Some biodiversity indicators at landscape scale could be derived from satellite remote sensing data such as forest cover, productivity and so on.

4) This protocol aims to monitor biodiversity of farm cluster. The pro

5) I suggest to compare this protocol with similar protocols in discussion mentioned in the paper like biodiversity indicators for organic and low-input farming systems (BioBio), Farm4Bio, and European Monitoring of Biodiversity in Agricultural Landscapes.

Reviewer #2: This research holds significant value for both ecological science and practical agricultural conservation in Europe. It directly addresses the critical challenge of farmland biodiversity loss by developing and rigorously evaluating a standardized, scalable monitoring protocol. The core innovation lies in its application within the "Farmer Cluster" model—a bottom-up, collaborative conservation approach. By creating a robust methodology to assess biodiversity outcomes across these clusters, the study provides an essential tool for empirically testing whether community-led agricultural initiatives can effectively reverse biodiversity decline at a meaningful landscape scale. The use of a Before-After-Control-Impact (BACI) experimental design significantly strengthens the potential to attribute observed changes in biodiversity to the interventions themselves, moving beyond mere observation to causal inference.

However, I do not believe this is a research article. It strikes me more as a research proposal or a work guideline. This depends on the journal's positioning. In terms of strengthening practical guidance and feasibility, greater consideration should be given to the following issues—although these may not necessarily need to be addressed within the scope of the present study, but rather as key priorities for the authors’ future follow-up research. From the perspective of the work itself, I think at least three additional points need to be considered in its actual implementation.

1. The monitoring methods, although well-documented, can be labor-intensive and require significant expertise, potentially limiting their long-term application outside well-funded research projects. The BACI design, while robust, also depends heavily on the availability of suitable control sites, which may be difficult to maintain over time due to changing land-use practices.

2. Integrate adaptive elements into the monitoring framework, allowing limited customization to regional conditions without compromising comparability. Incorporating remote sensing or automated monitoring tools could reduce costs and labor demands, making the protocol more scalable.

3. Stronger integration of socio-economic indicators—such as farmer engagement and management changes—would help link ecological outcomes more directly to decision-making processes within farmer clusters.

**Do you want your identity to be public for this peer review?** For information about this choice, including consent withdrawal, please see our Privacy Policy

Reviewer #1: No

Reviewer #2: No

---

## [Author Response · Author response to Decision Letter 1]

26 Feb 2026

Journal Requirements:

>> We have made edits to meet the style requirements.

“This Project has received funding from the European Union's Horizon 2020 research and innovation programme under grant agreement No 862731”

>> We have included the updated statement in our cover letter.

3. We note that you have referenced Nichols et al.” and “McHugh et al.” which has currently not yet been accepted for publication. Please remove this from your References and amend this to state in the body of your manuscript: “Nichols et al., unpublished” and “McHugh et al., unpublished” as detailed online in our guide for authors

>> These papers are now published, so the reference list and citations have been updated to include them correctly.

4. We note that Figures 1 and 2 in your submission contain map and satellite images which may be copyrighted. All PLOS content is published under the Creative Commons Attribution License (CC BY 4.0), which means that the manuscript, images, and Supporting Information files will be freely available online, and any third party is permitted to access, download, copy, distribute, and use these materials in any way, even commercially, with proper attribution.

For these reasons, we cannot publish previously copyrighted maps or satellite images created using proprietary data, such as Google software (Google Maps, Street View, and Earth). For more information, see our copyright guidelines: http://journals.plos.org/plosone/s/licenses-and-copyright.

1. You may seek permission from the original copyright holder of Figure(s) [#] to publish the content specifically under the CC BY 4.0 license.

>> Both figures have now been revised, and contain maps that do not require copyright statements, or that have been correctly cited and can be used.

5. We note that Figures 3 and s10 in your submission contain copyrighted images. All PLOS content is published under the Creative Commons Attribution License (CC BY 4.0), which means that the manuscript, images, and Supporting Information files will be freely available online, and any third party is permitted to access, download, copy, distribute, and use these materials in any way, even commercially, with proper attribution. For more information, see our copyright guidelines: http://journals.plos.org/plosone/s/licenses-and-copyright.

1. You may seek permission from the original copyright holder of Figure(s) [#] to publish the content specifically under the CC BY 4.0 license.

>> We do not believe that any of the photographs used in Figure 3 are under any copyright laws. They were all take by project partner members and have been provided for use in this paper. If one of these has been highlighted as copyright, please tell us which one and where it is published, as we are unaware that they have been published elsewhere.

S10 has now been removed from the submission.

Additional Editor Comments:

Please respond to the concerns of the referees and revised the manuscript accordingly

Reviewers' comments:

Reviewer's Responses to Questions

Comments to the Author

1. Is the manuscript technically sound, and do the data support the conclusions?

Reviewer #1: Yes

Reviewer #2: Partly

2. Has the statistical analysis been performed appropriately and rigorously?

Reviewer #1: N/A

Reviewer #2: I Don't Know

3. Have the authors made all data underlying the findings in their manuscript fully available?

Reviewer #1: Yes

Reviewer #2: No

4. Is the manuscript presented in an intelligible fashion and written in standard English?

Reviewer #1: Yes

Reviewer #2: Yes

5. Review Comments to the Author

Reviewer #1: Comments on MS PONE-D-25-36303 entitled “A standardised protocol for measuring farmland biodiversity outcomes across European Farmer Cluster landscapes” by Nichols et al.

1. General comments:

This paper proposes a standardized protocol for monitoring farmland biodiversity in European Farmer Cluster landscapes. The protocol is designed to evaluate and unify methods for assessing biodiversity across different farming systems. It provides practical guidance on selecting survey squares and transects, as well as detailed procedures for surveying birds, pollinators, and vegetation. Comprising a suite of standardized sub-protocols, the framework is adaptable to diverse agricultural contexts. Finally, the authors evaluate the effectiveness of these biodiversity indicators and survey methods. This paper is of good scientific quality, thus may interest the readers of PLoS One. However, there are some aspects that could be improved for me. Therefore, I recommend this paper to have a major revision. After this major revision, the paper may contribute to the field.

>> Thank you for your comments and providing suggestions for improvement.

2. Specific comments

1) Is it necessary to include some indicators of biodiversity pressures such as the annual application of pesticides, annual application of fertilizers, per capita annual net income of rural households and so on. These indicators could facilitate the subsequent analysis after biodiversity investigation.

2) Is it necessary to include some indicators of ecosystem services such as water purification, reducing pesticide needs and so on.

3) Some biodiversity indicators at landscape scale could be derived from satellite remote sensing data such as forest cover, productivity and so on.

>> Thank you for your suggestions. We agree that other environmental pressures are highly important, and were indeed considered. However, they were considered separately, as part of the Farmer Cluster sustainability analysis, and Farmland Ecosystem Assessment Support Tool (FEAST), so were not included in this methods protocol directly. We have added this text to the discussion:

L578-579: “Although other environmental indicators, services, and pressures were considered and are ecologically important…”

L588-594: “Additionally, environmental pressures such as pesticide use, fertiliser applications, water quality, and area of semi-natural habitat are all important indicators that were considered through a separate FC sustainability analysis (48), and ecosystem service indicators along with habitat suitability metrics were considered through the Farmland Ecosystem Assessment Support Tool (FEAST; (103)). These could be combined alongside the biodiversity indicators suggested above, should the FC have the funding and resources to include them.”

4) This protocol aims to monitor biodiversity of farm cluster. The pro

>> I emailed the editor to ask if the reviewer could send the full comment – I received the following text from the editor:

“The reviewer's comments are as follows:

The purpose of this protocol is to monitor biodiversity within farm clusters. It encompasses multiple taxon-specific subprotocols (e.g., for birds, insect pollinators, and vegetation). The current descriptions can be made more concise by increasing cross-references to these individual subprotocols.”

In response to this, we are struggling to see how the descriptions could be made more concise by cross-referencing. What the referee is proposing is unclear, and in the absence of an example, we don’t feel we can address this.

5) I suggest to compare this protocol with similar protocols in discussion mentioned in the paper like biodiversity indicators for organic and low-input farming systems (BioBio), Farm4Bio, and European Monitoring of Biodiversity in Agricultural Landscapes.

>> We have now added references of similar projects throughout the discussion, highlighting similarities and differences between our methods. Some examples:

L450-454: “The European Farmland Bird Indicator (EFBI) was created by Gregory et al. (51), and since then it has been used to predict that multiple European countries would witness a decline in farmland bird species in response to agricultural land-use change (52), making farmland birds a significant landscape-scale biodiversity indicator to monitor, hence their inclusion in many similar projects (16,24,25).”

L466-477: “Similar transect line surveys are already conducted as part of national bird census initiatives across Europe (56–58), with the data often used in large-scale biodiversity monitoring projects (20) and collated through the Pan-European Common Bird Monitoring Scheme, hence this method was chosen to be implemented in the FCs. However, some landscape-scale projects opt for different methods, such as “area searching” (mapping all birds that were seen or heard; (25)) or point counts, the latter with the potential to be used in place of transect walk. For example, the Zeeasterweg, Lelystad FC in the Netherlands conducted a point count for each transect/survey square to align more closely with their country’s national monitoring scheme (59). Additionally, the Bass-Durance FC in France began their breeding bird surveys earlier than the protocol specified to capture migratory species

---

## [Editor Report · Decision Letter 1]

9 Mar 2026

A standardised protocol for measuring farmland biodiversity outcomes across European Farmer Cluster landscapes

PONE-D-25-36303R1

Dear Dr. Nichols,

We’re pleased to inform you that your manuscript has been judged scientifically suitable for publication and will be formally accepted for publication once it meets all outstanding technical requirements.

Kind regards,

RunGuo Zang

Academic Editor

PLOS One

Additional Editor Comments (optional):

accept
---

## [Editor Report · Acceptance letter]

PONE-D-25-36303R1

PLOS One

Dear Dr. Nichols,

I'm pleased to inform you that your manuscript has been deemed suitable for publication in PLOS One. Congratulations! Your manuscript is now being handed over to our production team.

Kind regards,

on behalf of

Professor RunGuo Zang

Academic Editor

PLOS One